# Small RNA zippers lock miRNA molecules and block miRNA function in mammalian cells

Lingyu Meng[1,2,*], Cuicui Liu[1,2,*], Jinhui Lü[1,3,*], Qian Zhao[1], Shengqiong Deng[1], Guangxue Wang[1], Jing Qiao[1], Chuyi Zhang[1], Lixiao Zhen[1], Ying Lu[1], Wenshu Li[3], Yuzhen Zhang[1], Richard G. Pestell[4], Huiming Fan[1], Yi-Han Chen[1], Zhongmin Liu[1] & Zuoren Yu[1]

MicroRNAs (miRNAs) loss-of-function phenotypes are mainly induced by chemically modified antisense oligonucleotides. Here we develop an alternative inhibitor for miRNAs, termed 'small RNA zipper'. It is designed to connect miRNA molecules end to end, forming a DNA–RNA duplex through a complementary interaction with high affinity, high specificity and high stability. Two miRNAs, miR-221 and miR-17, are tested in human breast cancer cell lines, demonstrating the 70~90% knockdown of miRNA levels by 30–50 nM small RNA zippers. The miR-221 zipper shows capability in rescuing the expression of target genes of miR-221 and reversing the oncogenic function of miR-221 in breast cancer cells. In addition, we demonstrate that the miR-221 zipper attenuates doxorubicin resistance with higher efficiency than anti-miR-221 in human breast cancer cells. Taken together, small RNA zippers are a miRNA inhibitor, which can be used to induce miRNA loss-of-function phenotypes and validate miRNA target genes.

[1] Research Center for Translational Medicine, Key Laboratory of Arrhythmias of the Ministry of Education of China, East Hospital, Tongji University School of Medicine, 150 Jimo Road, Shanghai 200120, China. [2] East Hospital, Dalian Medical University, 150 Jimo Road, Shanghai 200120, China. [3] School of Basic Medical Sciences, Wenzhou Medical University, Wenzhou 325035, China. [4] Department of Cancer Biology, Sidney Kimmel Cancer Center, Thomas Jefferson University, Philadelphia, Pennsylvania 19107, USA. * These authors contributed equally to this work. Correspondence and requests for materials should be addressed to Z.Y. (email: zuoren.yu@tongji.edu.cn).

MicroRNAs (miRNAs) are a class of singled-stranded small RNA molecules that regulate the stability or translational efficiency of targeted messenger RNAs[1]. miRNAs are initially transcribed to primary miRNA, which are then cleaved by endonucleases to generate hairpin-structured precursor miRNAs (pre-miRNAs) with $60 \sim 70$ nucleotides in length. After transporting to the cytoplasm by Exportin-5, pre-miRNAs are processed by Dicer to generate mature miRNAs with $18 \sim 24$ nt in length[2,3]. More than 2,000 miRNA sequences have been identified or predicted from human origin tissues or cells[4]. miRNAs are mediated by RNA-induced silencing complex that lead to base-pairing interactions between an miRNA and the binding site of its target mRNAs mostly within the 3′-untranslated region (3′-UTR). miRNAs regulate diverse biological processes including cell fate determination, cell cycle progression, stem cell self-renewal, cancer initiation, progression and metastasis[5–11].

Small non-coding RNA-based diagnostic and therapeutic applications for human cancer are expected in the near future. Synthetic miRNA mimics or miRNA expression vectors carrying either a pre-miRNA sequence or an artificial miRNA hairpin sequence have been successfully applied to restore or overexpress miRNA in vitro. Chemically modified antisense oligonucleotides are widely used to knock down miRNA. In terms of the oncogenic function of a subset of miRNAs, such as miR-21 in breast cancer[12], decreasing the expression level or blocking the function of these miRNAs is believed to be a promising strategy for cancer treatment. Currently, chemically modified miRNA inhibitors are frequently applied. The modification includes addition of 2′-O-methyl, 2′-O-methoxyethyl, locked nucleic acid (LNA) with 2′-O connecting to 4′-C and so on. Although the modified nucleic acid structure has high affinity to bind with a target miRNA, the loss-of-function studies of miRNAs still need to be scrutinized, to improve the binding specificity and exclude the frequent off-target effects. In addition, two other approaches were reported to block miRNA function. One is called miRNA-sponge[13], which serves as a competitive inhibitor of miRNAs. An expression vector carrying multiple binding sites to an miRNA is introduced into cells. Following the vector gene transcription, the overexpressed synthetic binding sequences occupy the endogenous miRNA in the cells with high-affinity blocking miRNA regulation of its target genes. The other approach is called miRNA mask[14], which uses oligonucleotides perfectly complementary to miRNA-binding sites of target mRNAs. The miRNA mask blocks the access of a miRNA to the binding sequence of target mRNAs, thereby blocking the miRNA–mRNA interaction. However, the approaches for miRNA loss of function using either miRNA sponge or miRNA mask are not widely accepted due to the limited effect and complicated regulatory mechanisms in vivo.

Breast cancer is the most common cancer in women. Although the 5-year survival has reached around 98% in localized breast cancer[15], metastatic breast cancer, which occurs in 20–30% of breast cancer patients, remains incurable due to the lack of a reliable target for treatment[16]. miRNAs regulate breast cancer initiation and progression[3,9]. Altered expression of miRNAs or mutation of miRNA genes have been described in human breast cancer[9,17]. Zhang et al.[18] analysed 283 human miRNA genes in 55 human breast primary tumours and 18 human breast cancer cell lines, demonstrating a high frequency ($\sim 72.8\%$) of gene copy number abnormality in miRNA-containing regions in human breast cancer. A growing body of evidence implicates miRNAs in regulating breast cancer metastasis and epithelial to mesenchymal transition[19,20].

Herein, DNA oligos are specially designed to bind to the 5′-half sequence of one molecule and the 3′-half sequence of another

molecule of the target miRNA through a complementary interaction, which we term a 'small RNA zipper'. As the small RNA zippers are chemically modified as LNAs, the binding to target miRNA has high affinity, high specificity and high stability. The small RNA zipper is able to connect the target miRNA molecules end to end forming a stable structure, blocking the functions of the miRNA. Two miRNAs, miR-221 and miR-17, has been validated in human breast cancer cell lines demonstrating $\sim 70$–90% knockdown of miRNA levels by corresponding small RNA zippers at 24 h after transfection. The miR-221 zipper can rescue the expression of p27, a target gene of miR-221, and reverse the cell-migration-promoting function of miR-221 in breast cancer cells. In addition, by application of the miR-221 zipper, we demonstrate that miR-221 may confer doxorubicin (Dox) resistance in human breast cancer cells.

## Results

**Construction of small RNA zippers.** In view of the small size of miRNAs, an oligonucleotide was specially designed to be complementary to the second half sequence of a miRNA molecule and the first half of another, thereafter connecting miRNA molecules end to end as shown in Fig. 1a. It is named a small RNA zipper, in which a nucleotide gap was designed between two miRNA molecules to provide space for a stable structure formation to ensure binding specificity to its target sequence. As such, for a given miRNA, all individual molecules can be connected end to end via the small RNA zipper following the zipper strategy (Fig. 1b). To avoid self-complementarity and to enhance binding specificity, LNA nucleosides were applied to synthesize the small RNA zipper. LNA is a class of nucleic acid analogues in which the ribose ring is 'locked' by a methylene bridge connecting the 2′-O atom and the 4′-C atom.

**Effect of small RNA zippers on target miRNA levels.** To determine the effect of the small RNA zipper on the target miRNA level, miR-17 zipper was synthesized (Fig. 2a) and transfected into breast cancer cell lines MCF-7 and MDA-MB-231. The abundance of miR-17 in the cells was analysed and compared using quantitative real-time PCR before and after transfection of the miR-17 zipper. miR-17 abundance was knocked down around 70–90% by miR-17 zipper at 24 and 48 h after transfection, which was confirmed by quantitative real-time PCR using both the SYBR approach (Fig. 2b,c) and the Taqman approach (Supplementary Fig. 1A). In comparison, a negative control zipper did not show an effect on the miR-17 abundance (Supplementary Fig. 1B).

In addition to miR-17, miR-221, which has been reported to promote cellular migration and invasion in breast cancer[4], was tested by small RNA zipper (Fig. 2d–g). miR-221 has high expression in the MDA-MB-231 breast cancer cell line but

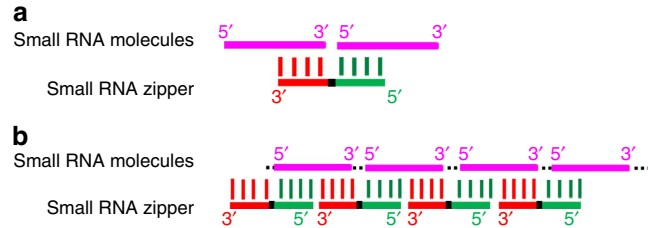

**Figure 1 | The construction of small RNA zipper.** (**a**) Schematic representation of the interaction between small RNA zipper and small RNAs. (**b**) Schematic representation of how the small RNA zippers connect the target small RNA molecules end to end, forming an RNA–DNA duplex.

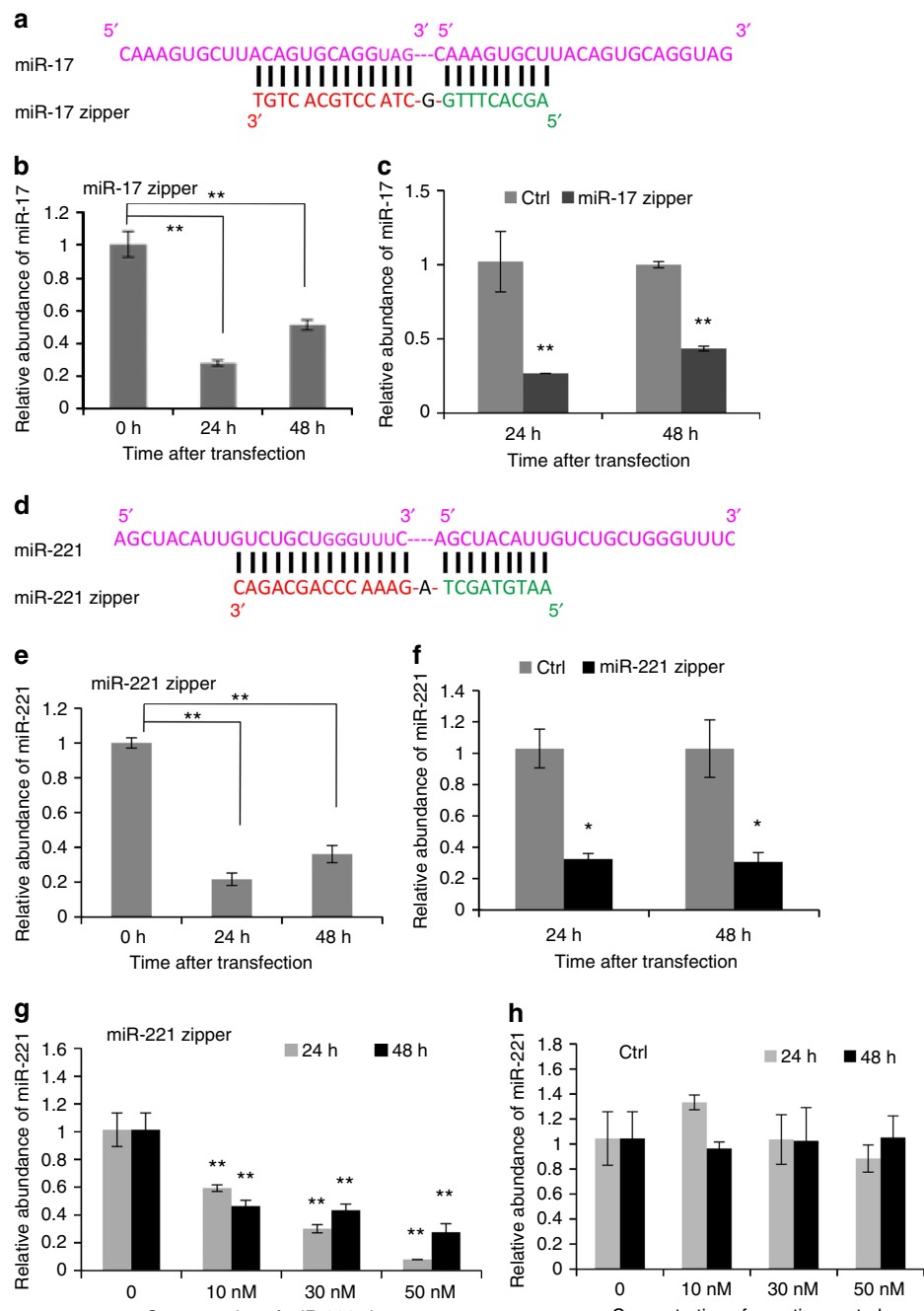

**Figure 2 | The effect of small RNA zippers on the target miRNA levels.** (**a**) Construction of miR-17 zipper. (**b,c**) Quantitative real-time PCR analysis showing the decreased level of miR-17 in MDA-MB-231 cells after transfection of miR-17 zipper. Data are mean ± s.e.m. **$P < 0.01$ (miR zipper groups versus control groups or 0 h before transfection, t-test, $n = 3$). (**d**) Construction of miR-221 zipper. (**e,f**) Quantitative real-time PCR analysis showing the decreased level of miR-221 in MDA-MB-231 cells after transfection of miR-221 zipper. Data are mean ± s.e.m. *$P < 0.05$ and **$P < 0.01$ (miR zipper groups versus control groups or 0 h before transfection, t-test, $n = 3$). (**g,h**) Quantitative real-time PCR analysis showing the dose response of the miR-221 levels to the concentration of the miR-221 zipper (**g**) and negative control (**h**), demonstrating the highest efficiency under the condition of 50 nM concentration and 24 h transfection. Data are mean ± s.e.m. **$P < 0.01$ (different concentrations of miR zipper groups versus 0 nM blank control, t-test, $n = 3$).

low expression in MCF-7 cells; therefore, MDA-MB-231 cells were used to test the effect of the miR-221 zipper. Similar to the miR-17 zipper, the miR-221 zipper decreased the amount of miR-221 at 24 and 48 h after transfection compared with a negative control (Fig. 2e,f and Supplementary Fig. 2A). A dose response assay indicated the knockdown efficiency of miR-221 was positively correlated with the concentration of the miR-221 zipper (Fig. 2g and Supplementary Fig. 2). The control zipper

did not affect the expression of miR-221 up to a high concentration of 50 nM (Fig. 2h). miR-221 expression was knocked down ∼90% by the miR-221 zipper at the concentration of 50 nM in MDA-MB-231 cells, which showed much higher efficiency than the lower concentrations. In contrast, miR-222, another member of the miR-221/222 cluster but with a different sequence to miR-221 (Supplementary Fig. 3A), did not show a knockdown effect by the miR-221 zipper

(Supplementary Fig. 3B), indicating the sequence specificity of small RNA zipper binding to target miRNAs. In addition, a head-to-head comparison between miR-221 zipper and anti-miR-221 indicated that both approaches can knockdown miR-221 around 70–90%, depending on the concentration and transfection efficiency (Supplementary Fig. 4).

**Optimization of small RNA zipper construction.** As shown in Fig. 1, small RNA zipper technology was developed to connect the target small RNA one by one. In consideration of the RNA–DNA duplex formation and sequence-specific binding, one additional nucleotide was designed in the zipper sequence, to introduce a nucleotide gap between two target small RNAs. To demonstrate whether the gap is required, a LNA oligonucleotide without a gap between two target RNA molecules, named small RNA Δzipper, was designed for comparison. Following this strategy, the LNA miR-221 Δzipper and the miR-17 Δzipper were synthesized as shown in Fig. 3. The efficiency of miR-221 knockdown was compared between the same concentrations of miR-221 zipper and Δzipper (Fig. 3a–d). Both miR-221 zipper and Δzipper decreased the abundance of miR-221 in MDA-MB-231 cells, whereas miR-221 zipper showed a higher efficiency than miR-221 Δzipper (Fig. 3b–d).

The comparison between the miR-17 zipper and Δzipper (Fig. 3e) was performed in MCF-7 cells. The abundance of miR-17 was analysed using northern blot hybridization (Fig. 3f). The effect of the miR-17 zipper on miR-17 abundance was dose dependent. miR-17 zipper knocked down the miR-17 level at the concentrations of 20 and 30 nM, whereas miR-17 Δzipper

was not as effective as miR-17 zipper at the concentration of 20 nM, indicating the higher efficiency of the miR-17 zipper than miR-17 Δzipper to reduce the miR-17 level.

**Sequence specificity of small RNA zipper.** To validate the sequence specificity of binding between a small RNA zipper and target miRNA, miR-222 was tested in MDA-MB-231 cells transfected with miR-221 zipper. miR-221 and miR-222, two members of a miRNA cluster, which is located on chromosome X, share the same 'seed' sequence but the rest of the sequence is different (Supplementary Fig. 3A). As described above, knockdown of miR-221 abundance showed a dose response to the miR-221 zipper (Fig. 2g). However, miR-222 did not show a response to either low or high concentrations of miR-221 zipper (Supplementary Fig. 3B). This was further confirmed when the miR-221 zipper and Δzipper were applied, to determine the effect on the abundance of miR-221 and miR-222 (Fig. 3c,d).

Let-7 was tested to further validate the specificity of the miRNA zipper because of the strong homology between let-7 family members. The family members including let-7a, let-7b and let-7c have moderate expression in MDA-MB-231 cells (Supplementary Fig. 5). Let-7a zipper (Fig. 4a) was transfected to MDA-MB-231 cells followed by a quantitative analysis of let-7a, let-7b and let-7c in 24 h. It was clearly shown there was significant knockdown of let-7a, but not let-7b nor let -7c, by the let-7a zipper (Fig. 4b).

Point mutations including a two nucleotides mutation (miR-17 zipper mu1) and a single-nucleotide mutation (miR-17 zipper mu2)

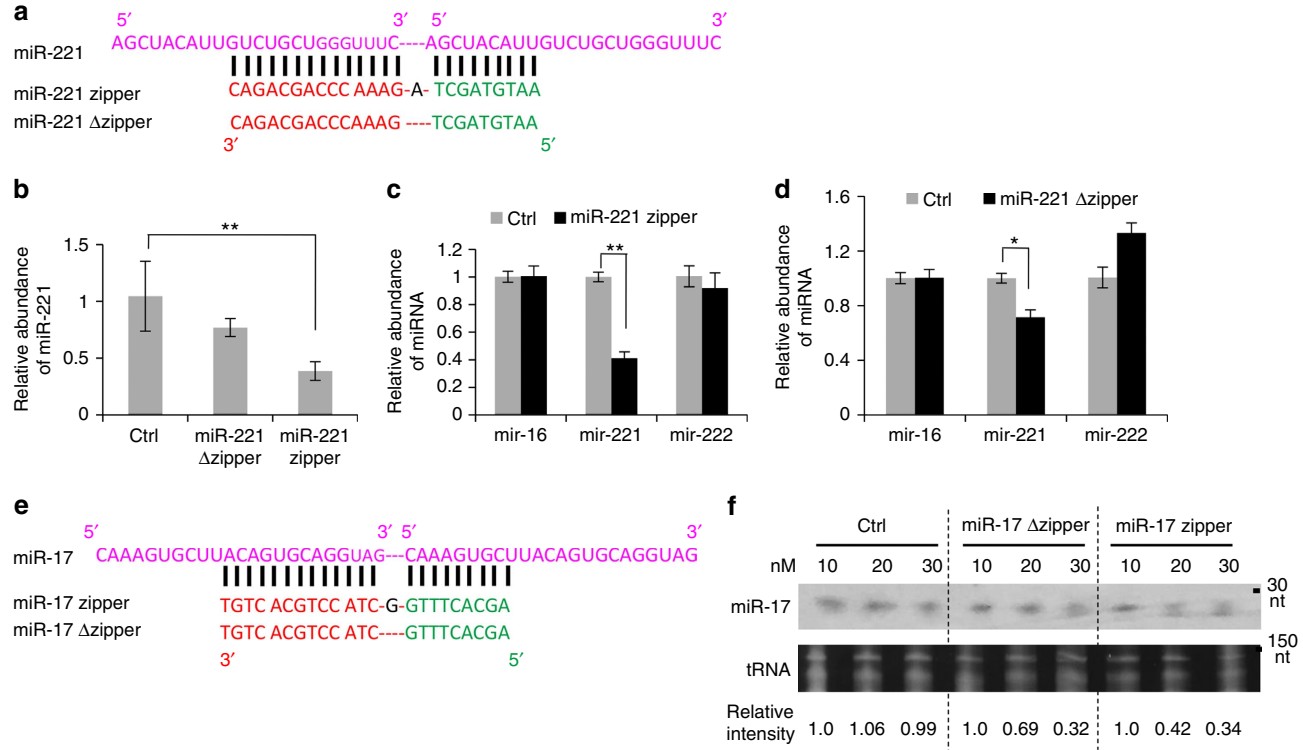

**Figure 3 | Comparison between miRNA zipper and miRNA Δzipper.** (**a**) Sequence and construction of miR-221 zipper and miR-221 Δzipper. (**b**) Quantitative real-time PCR analysis showing the different levels of miR-221 in MDA-MB-231 at 24 h after transfection of miR-221 zipper or miR-221 Δzipper or negative control. Data are mean ± s.e.m. **$P < 0.01$ (miR zipper group versus control group, $t$-test, $n = 3$). (**c,d**) Quantitative real-time PCR analysis showing the levels of miR-221, miR-222 and miR-16 in MDA-MB-231 at 24 h after transfection of miR-221 zipper (**c**) or miR-221 Δzipper(**d**). Data are mean ± s.e.m. *$P < 0.05$ and **$P < 0.01$ (zipper groups versus control groups, $t$-test, $n = 3$). (**e**) Sequence and construction of miR-17 zipper and miR-17 Δzipper. (**f**) Dose response validation of miR-17 levels in MCF-7 cells after transfection with different concentrations (10, 20 or 30 nM) of miR-17 zipper, miR-17 Δzipper and negative control, respectively. Northern blotting was performed. tRNA served as RNA loading control. See also Supplementary Fig. 8.

were applied to miR-17 zipper as shown in Fig. 4c. Consistently, miR-17 expression in MDA-MB-231 cells was knocked down 70~90% at 24 and 48 h after transfection with miR-17 zipper, but not with mutated miR-17 zippers (Fig. 4d). Notably, miR-17 zipper mu2 showed a small effect on the miR-17 expression, although the P-value did not support a significant change (Fig. 4d).

The C–G change of the miR-17 zipper mu2 was initially designed to producing a G–U wobble base pairing, to determine the effect of wobble pairing on the interaction between the miRNA zippers and miRNAs. The data demonstrated the limited effect of wobble pairing, further demonstrating the sequence specificity of a miRNA zipper interacting with target miRNA.

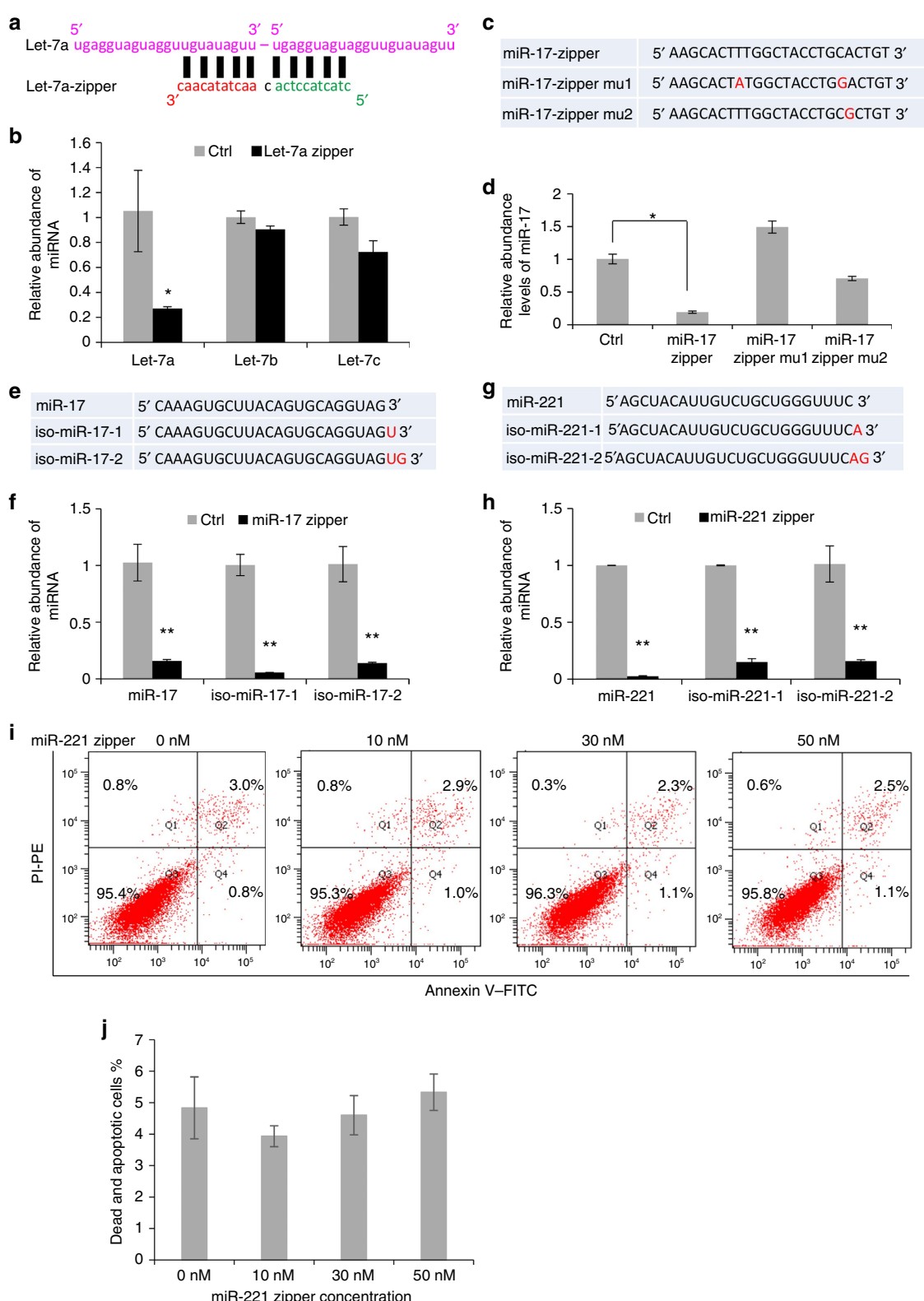

Emerging evidence indicates that many miRNAs have iso-miRs differing by ~1–3 nucleotides at either the 5′- or 3′-end of the miRNA molecules. To determine the effect of miRNA zippers on iso-miRs, quantitative analysis of miR-17, iso-miR-17-1 (one additional nucleotide at the 3′-end) and iso-miR-17-2 (two additional nucleotides at the 3′-end; Fig. 4e) were performed in MDA-MB-231 cells following miR-17 zipper treatment (Fig. 4f). Both iso-miR-17-1 and iso-miR-17-2 themselves have lower expression than miR-17 in cells (Supplementary Fig. 6A). Our analysis indicated that the effect of the miR-17 zipper was not limited to miR-17, but also to the expression of two iso-miRs (Fig. 4e,f). Similarly, the miR-221 zipper was demonstrated to be able to knockdown both miR-221 and iso-miR-221 in cells (Fig. 4g,h). As the complementary binding between the miRNA zipper and iso-miRs is totally the same as that between the miRNA zipper and miRNA, the interaction between the miRNA zipper and iso-miRs may follow the structure as shown in Supplementary Fig. 6B. In consideration of isomiRs with three or more extra nucleotides, which have not been tested yet by the miRNA zippers, isomiR-sequence-specific zipper is strongly suggested rather than miR zipper, to knock down isomiRs.

**Functional validation of small RNA zipper.** To determine whether the small RNA zippers are safe and functional in cells, different concentrations (0, 10, 30 and 50 nM) of miR-221 zippers were introduced into MDA-MB-231 cells. After 24 h, Annexin V staining was performed to quantify the dead and apoptotic cells (Fig. 4i). Quantitative analysis indicated ~5% of total dead and apoptotic cells in all tests. No significant difference was found between control and miR-221 zipper-treated cells (Fig. 4j), indicting the non-toxicity of small RNA zipper to cells. $H_2O_2$ was used as a positive control to induce cell apoptosis (Supplementary Fig. 7).

Anti-miRNA has been shown to knock down target miRNA and thereby block its biological function. miRNA zippers have similar effects on the target miRNA levels. The cellular phenotypes of the miR-221 zipper and anti-miR-221 in MDA-MB-231 cells were compared. It has been well demonstrated that miR-221 promotes cell migration and invasion in MDA-MB-231 cells[4], which was attenuated by transfection of anti-miR-221 as shown in Fig. 5a,b. Similarly, transfection of miR-221 zipper into MDA-MB-231 cells suppressed the cellular migration (Fig. 5c). The migration of living cells was recorded for 48 h (Supplementary Movie 1 and 2, and Fig. 5c) and quantitatively analysed (Fig. 5d). At the same time, the effect on the cellular migration of miR-221 Δzipper was tested and compared with miR-221 zipper. miR-221 Δzipper showed an inhibitory function that was less than miR-221 zipper (Fig. 5c,d and Supplementary Movie 3).

miR-221 has been reported to promote tamoxifen resistance in ERα-positive breast cancer cells[21,22]. We tested whether miR-221 has any correlation with Dox resistance in breast cancer cells. Surprisingly, miR-221 expression showed a ~5-fold increase in the Dox-resistant breast cancer cell lines including MCF-7 (Fig. 5e) and MDA-MB-231 (Fig. 5f). Introduction of anti-miR-221 into MDA-MB-231 cells increased the sensitivity to Dox (Fig. 5g). Similarly, transfection of the miR-221 zipper into MDA-MB-231 cells attenuated Dox resistance as shown in Fig. 5h. Notably, the miR-221 zipper was more effective than anti-miR-221, to increase cellular sensitivity to 200 nM Dox (Fig. 5g,h).

**Effect of small RNA zipper on target genes of miRNA.** To determine the effects of the miRNA zipper on the expression of miRNA target genes, the interaction between miR-221 and *p27* was assessed on application of the miR-221 zipper. *p27* mRNA carries two binding sites for miR-221/222 in the 3′-UTR (Fig. 6a). *p27* has been well confirmed to be a target gene of miR-221/222 in breast cancer. Consistent with the knockdown of miR-221, *p27* expression was upregulated by the miR-221 zipper in MDA-MB-231 cells (Fig. 6b,c). The luciferase reporter constructs carrying either wild-type (WT) *p27* 3′-UTR or point mutated *p27* 3′-UTR (mutations in the two binding sites of miR-221/222) were tested to interact with miR-221/222 in MDA-MB-231 cells (Fig. 6d). As expected, increased luciferase activity was observed with miR-221 zipper applied to WT *p27* 3′-UTR (Fig. 6e). Owing to the unaffected expression of miR-222 in the miR-221 zipper-treated cells, the increased level of luciferase activity by miR-221 zipper was not as high as that by mutated *p27* 3′-UTR. (Fig. 6e).

**Discussion**

Since the first discovery of miRNA in 1993 (ref. 23) and the confirmation of miRNA function in 2001 (refs 24–26), thousands of miRNAs have been identified from eukaryotes. The aberrant expression of miRNA has been frequently reported in various diseases including cancer[9]. However, the function of most miRNAs remains to be discovered and few miRNA–target interactions have been experimentally validated *in vitro* or *in vivo*. To identify the function of miRNA and determine the relationship of miRNAs to diseases, different approaches have been applied to result in a loss or gain of miRNA function accompanied with increased or decreased levels of endogenous target genes. Presently, loss-of-function phenotypes are induced by means of chemically modified antisense oligonucleotides including 2′O-methyl, LNA, phosphorothioate, cholesterol and so on, which work by annealing to mature miRNAs to block miRNA from binding to target genes. Herein, a small RNA zipper

**Figure 4 | Specificity of miRNA zipper to knock down target miRNA.** (**a**) Sequence and construction of miRNA let-7a zipper. (**b**) Quantitative analysis of let-7a, let-7b and let-7c in MDA-MB-231 cells at 24 h after transfection with 30 nM let-7a zipper. Data are mean ± s.e.m. *P < 0.05, (zipper group versus control group, t-test, n = 3). (**c**) Sequences of miR-17-zipper and point mutated miR-17 zippers (miR-17-zipper mu1 and mu2). The mutated nucleotides were indicated with red font. (**d**) Quantitative analysis of the miR-17 levels in MDA-MB-231 cells at 24 h after transfection with 30 nM of control, mR-17-zipper and mutated miR-17-zippers, respectively. Data are mean ± s.e.m. *P < 0.05 (zipper groups versus control group, t-test, n = 3). (**e**) Sequences of mature miR-17 and two miR-17 iso-miRs which have one to two additional nucleotides at the 3′-end of miR-17. (**f**) Quantitative analysis of miR-17 and iso-miRs in MDA-MB-231 cells at 24 h after transfection with 30 nM of mR-17-zipper. The results indicated the knockdown effects of both miR-17 and iso-miRs by miR-17 zipper. Data are mean ± s.e.m. **P < 0.01 (zipper groups versus control groups, t-test, n = 3). (**g**) Sequences of mature miR-221 and miR-221 iso-miRs which have 1-2 additional nts at the 3′ end of miR-221. (**h**) Quantitative analysis of miR-221 and iso-miRs in MDA-MB-231 cells at 24 h after transfection with 30 nM of mR-221-zipper. The results indicated the knockdown effects of both miR-221 and iso-miR-221 by miR-221 zipper. Data are mean ± s.e.m. **P < 0.01 (zipper groups versus control groups, t-test, n = 3). (**i**) Annexin V staining for dead and apoptotic MDA-MB-231 cells at 24 h after transfection with 0, 10, 30 and 50 nM of miR-221 zipper, respectively. (**j**) Quantitative analysis of I indicated ~5% of total dead and apoptotic cells in all groups. No significant difference for dead and apoptotic cell percentage was found between cells treated with different concentrations of miR-221 zipper (t-test, n = 3).

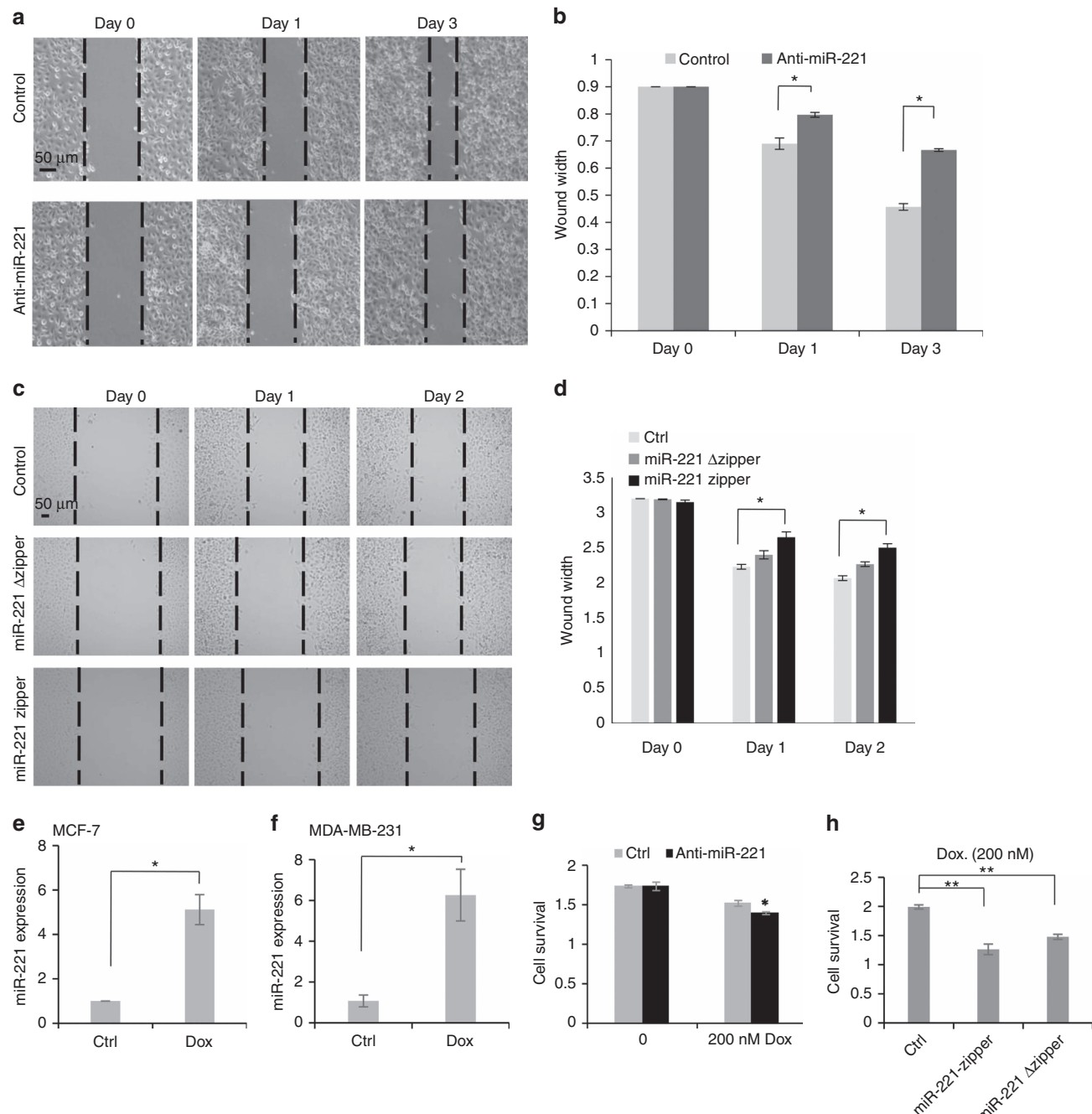

**Figure 5 | Functional validation of miR-221 zipper in human breast cancer cells.** (**a**) Anti-miR-221 suppressed cellular migration in MDA-MB-231 cells. Scale bar, 50 μm. (**b**): Quantitative analysis of A. Data are mean ± s.e.m. *$P < 0.05$ (anti-miR-221 groups versus control groups, t-test, $n = 3$). (**c**) The effects of miR-221 zipper and miR-221 Δzipper on cellular migration after transfection into MDA-MB-231 cells. The migration of living cells was recorded continuously for 48 h. Scale bar, 50 μm. (**d**) Quantitative analysis of C. Data are mean ± s.e.m. *$P < 0.05$ (zipper groups versus control groups, t-test, $n = 3$). (**e**) Increased miR-221 expression in Dox-resistant MCF-7 cells. Data are mean ± s.e.m. *$P < 0.05$ (Dox. group versus control group, t-test, $n = 3$). (**f**) Increased miR-221 expression in Dox-resistant MDA-MB-231 cells. Data are mean ± s.e.m. *$P < 0.05$ (Dox. group versus control group, t-test, $n = 3$). (**g**) anti-miR-221 promoted the sensitivity of MDA-MB-231 cells to 200 nM Dox. Data are mean ± s.e.m. *$P < 0.05$ (anti-miR-221 group versus control group, t-test, $n = 3$). (**h**) miR-221 zipper promoted the sensitivity of MDA-MB-231 cells to 200 nM Dox. Data are mean ± s.e.m. **$P < 0.01$ (zipper groups versus control group, t-test, $n = 3$).

was specifically designed and successfully applied to connect miRNA molecules end by end, forming a DNA–RNA duplex through a complementary interaction with high affinity, high specificity and high stability. Two miRNAs, miR-221 and miR-17, were tested in human breast cancer cell lines, demonstrating a dose-dependent knockdown efficiency of

miR-221 and miR-17 by the miR-221 zipper and miR-17 zipper, respectively. With up to 50 nM concentrations of the miRNA zippers, the expression level of target miRNA can be suppressed around 70~90%. The specificity of miRNA zippers was confirmed by testing homologous miRNA family members and miRNA zipper point mutations. A head to

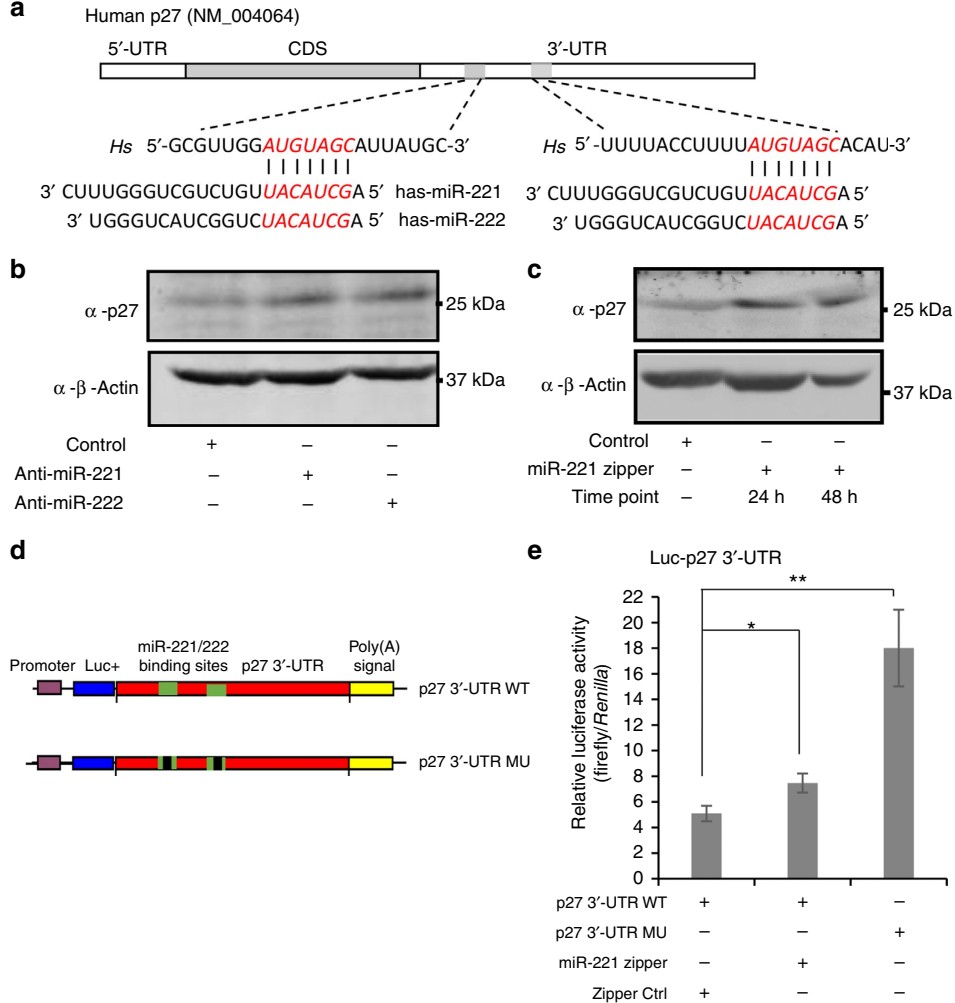

**Figure 6 | Effect of miR-221 zipper to the target gene of miR-221.** (**a**) Sequence alignment showing the 3′-UTR of human *p27* mRNA has two binding sites to miR-221/222. (**b**) Western blot analysis showing increased *p27* expression by anti-miR-221 and anti-miR-222 in MDA-MB-231 cells. β-Actin served as protein loading control. See also Supplementary Fig.9. (**c**) Western blot analysis showing increased *p27* expression in MDA-MB-231 cells by miR-221 zipper at 24 and 48 h after transfection. β-Actin served as protein loading control. See also Supplementary Fig.10. (**d**) Schematic representation of the luciferase reporter constructs carrying either *p27* 3′-UTR WT or *p27* 3′-UTR MU (point mutated in the two binding sites of miR-221/222) right downstream of the luciferase coding region. (**e**) Luciferase reporter assay showed increased luciferase activity by miR-221 zipper to WT *p27* 3′-UTR. The increased level of luciferase activity was not as high as that in mutated *p27* 3′-UTR cells mainly due to the unaffected expression of miR-222 in miR-221 zipper treated cells. The firefly luciferase activity was normalized to *Renilla*. Values are presented as the mean ± s.e.m. ($n = 3$). *$P < 0.05$, **$P < 0.01$ (miR zipper group versus control group, p27 3′-UTR mutation group versus p27 3′-UTR WT group, *t*-test, $n = 3$).

head comparison between the miRNA zipper and antisense inhibitor indicated similar efficiency to knock down the target miRNA expression.

The miRNA zippers were confirmed to be non-toxic to cells. Transfection with 50 nM miRNA zipper to a human breast cancer cell line led to ~90% knockdown of most of the target miRNAs without inducing cell apoptosis. Furthermore, the miRNA zipper is functional in cells to reverse the cellular phenotypes caused by a given miRNA. For example, the miR-221 zipper rescued the suppression of the target gene *p27* by miR-221. The oncogenic function of miR-221, including promoting cellular migration and invasion in breast cancer cells, was reversed by the miR-221 zipper.

Taken together, these findings show the small RNA zipper is a type of inhibitor for small RNAs including miRNAs, which can induce miRNA loss-of-function phenotype. This may provide an approach to determine the function of a miRNA and also to confirm the target genes of miRNAs.

In view of the similar characteristics, including small size and single strand, shared by miRNAs, small interfering RNAs (siRNAs) and piwi-interacting RNAs (piRNAs), small RNA zipper technology should be applicable to induce loss-of-function of not only miRNAs but also endogenous siRNAs and piRNAs. That is why we have used term the small RNA zipper rather than miRNA zipper. However, it will clearly be necessary to experimentally validate the effect and function of small RNA zippers on other types of small RNAs including siRNAs and piRNAs.

To obtain a stable DNA–RNA duplex, LNA-enhanced oligonucleotides are applied to small RNA zippers, in which the LNA nucleic acid analogue is 'locked' by a methylene bridge connecting the 2′-O atom and the 4′-C atom. LNA can bind very tightly to complementary residues with high binding specificity, whereas avoiding unacceptable secondary structure and self-complementarity[27]. In addition, we have demonstrated that LNA G–T may not have wobble interaction as G–U by

producing a G–U wobble base pairing between the miRNA zippers and miRNAs.

By application of the miR-221 zipper in human breast cancer cells, a function of miR-221 was confirmed that high expression of miR-221 may be responsible at least in part for the Dox-resistance in breast cancer cells. miR-221 has been well demonstrated to promote cellular migration and invasion in breast cancer[4]. miR-221 was reported to promote tamoxifen resistance in ERα-positive breast cancer cells[22]. In the current study we found overexpression of miR-221 in the Dox-resistant breast cancer cells. Knockdown of miR-221 by miR-221 zipper increased the sensitivity of MDA-MB-231 cells to Dox. These observations indicate miR-221 may be a promising therapeutic target to prevent chemo resistance in breast cancer and prevent cancer cell metastasis.

## Methods

**Cell lines and cell culture.** Human breast cancer cell lines MDA-MB-231 and MCF-7 were originally purchased from ATCC (Manassas, VA) and maintained in our laboratory. DMEM medium containing penicillin and streptomycin (100 mg l$^{-1}$) and 10% fetal bovine serum (FBS) at 37 °C in a humidified environment with 5% $CO_2$ was applied for cell culturing.

**Oligos and transfection.** LNA miRNA zipper and miRNA Δzipper were designed following LNA Oligo Tools and Design Guidelines of Exiqon (Vedbaek, Denmark), and synthesized by Exiqon and GenScript (Nanjing, China). The oligo sequences are (from 5′ to 3′): miR-17 zipper 5′-A + AGCACTTTGGCTACCTGCACT + GT-3′; miR-17 Δzipper 5′-A + AGCACTTTGCTACCTGCACT + GT; miR-221 zipper 5′-AA + TGTAGCTAGAAACCCAGCA + GAC-3′; miR-221 Δzipper 5′-AA + TGTAGCTGAAACCCAGCA + GAC-3′; let-7a zipper 5′-CTACTA CCTCACAACT + ATACA + AC-3′; miR-17 zipper mu1 5′-A + AGCACTATG GCTACCTGGACT + GT-3′; and miR-17 zipper mu2 5′-A + AGCACTTT GGCTACCTGCGCT + GT-3′. Negative control was a commercially available LNA product from Exiqon. Anti-miR-221 and control were purchased from Ambion of Life Technologies. The HiPerFect transfection reagent from Qiagen was used for cell transfection following the manufacturer's instructions. Final concentration for miRNA zipper and anti-miRNA was 30 nM, except for dose–response assays.

**miRNA real-time PCR analysis.** Total RNA was extracted with Trizol reagent (Invitrogen). First-strand complementary DNA of miRNAs was prepared using the M&G miRNA Reverse Transcription kit (miRGenes, Shanghai, China) following the manufacturer's instruction. Forward primer sequences for real-time PCR of miRNAs were: miR-16, 5′-agcagcacgtaaatattggc-3′; miR-221, 5′-agctacattgtctgct-3′; miR-222, 5′-agctacatctggctact-3′; let-7a, 5′-tgaggtagtaggttgtata-3′; let-7b, 5′-tgagg-tagtaggttgtgtg-3′; let-7c, 5′-tgaggtagtaggtttgtatg-3′; iso-miR-17-1, 5′-tgcttacagtg caggtagt-3′; iso-miR-17-2, 5′-tgcttacagtgcaggtagtg-3′; iso-miR-221-1, 5′-ctacattgt ctgctgggtttca-3′; iso-miR-221-2, 5′-ctacattgtctgctgggtttcag-3′; and 5s ribosomal RNA, 5′-agtacttggatgggagaccg-3′. All the primer oligoes were synthesized and purified by GenScript. The Taqman probe was from Ambion. The SYBR Green Master Mix was ABI product (Applied Biosystem, Life Technologies). The ABI 7900 HT Sequence Detection System (Applied Biosystem, Life Technologies) was used for quantitative real time PCR assay. miR-16 and 5s rRNA were used for normalization in the SYBR approach. U6 was used for normalization in the Taqman approach

**Northern blot analysis.** Northern blot analysis of miRNAs was performed as described before[28]. Briefly, total RNA (10–20 μg per lane) was loaded on a 15% denaturing polyacrylamide gel and electrophoresed at 200 V until the bromophenol blue approached the bottom. The RNA was transferred from the gel to Hybond-N + membrane using a Semi-Dry Transfer Apparatus. DNA oligonucleotide probes were 5′-end labelled with [γ-32P]-ATP and hybridization was carried out using Rapid-Hyb buffer following the manufacturer's instructions (Amersham, Piscataway, NJ). Uncropped versions of all blots are available in Supplementary Figs 8 and 9.

**Western blot analysis.** Cell lysates (50 μg) were prepared after 24 and/or 48 h transfection of miRNA zipper or anti-miRNA and separated by 10% SDS–PAGE. The proteins were transferred to nitrocellulose membrane. After being blocked in 5% milk (w/v) at room temperature for 1 h, the membranes were incubated at 4 °C overnight with primary antibodies (1:2,000). Following 1 × PBST washing, the membranes were incubated with secondary antibodies (1:3,000) at room temperature for 1 h followed by staining. The following antibodies were used for western blotting: anti-p27 (sc-776) and anti-β-actin (sc-47778; purchased from

Santa Cruz Biotechnology). Uncropped versions of all blots are available in Supplementary Figs 8 and 9.

**Wound-healing assay.** Cells were counted and plated in equal numbers in 12-well tissue culture plates to achieve 95% confluence. Thereafter, a vertical wound was created using a 0.1 μl pipette tip. The cells were cultured with FBS-reduced DMEM medium (0.1% FBS). Images of the wound were captured at designated times to assess wound closure rate.

**Video recording of living cell migration.** MDA-MB-231 cells were seeded in a six-well plate, and growing until forming confluent monolayer. A ~300 μm-wide linear wound was made using a sterile 10 μl pipette tip. The suspend cells were washed away with PBS buffer. The remaining cells were cultured in DMEM medium containing 0.1% FBS. Cells were allowed to close the wound for 48 h and the wound areas were photographed with the Live Cell Imaging System (Leica DMI600B) every 15 min.

**Statistical analysis.** All data are presented as mean ± s.e.m. unless stated otherwise. The two-tailed Student's t-test (unpaired) was used to compare difference between groups, in which $P < 0.05$ was considered statistically significant.

**Data availability.** The data that support the findings of this study are available from the corresponding author upon reasonable request.

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

## Acknowledgements

This work was supported by the National Key Research and Development Program of China Stem Cell and Translational Research (2016YFA0101202); Natural Science Foundation of China (Grant 81572593); partly by grants 13JC1401702 and 124119a7100 from Science and Technology Commission of Shanghai Municipality; partly by NIH grants R01CA070896, R01CA075503, R01CA132115, R01CA107382 and, R01CA086072 (R.G.P.); generous grants from the Dr Ralph and Marian C. Falk Medical Research Trust (R.G.P.); and a grant from Pennsylvania Department of Health (R.G.P.). We thank Dr Reuven Agami for providing WT and mutated pGL3-p27 3′-UTR vector. We thank Dr Brian Tomlinson for providing English editing.

## Author contributions

Y.Z., L.Z., C.Y.H. and Z.Y. designed the project and wrote the paper. F.H. and R.G.P. did paper revisions and language editing. W.L. and Q.Z. analysed the data. L.M., L.C., J.L., S.D., G.W., J.Q., C.Z., Z.L. and Y.L. performed the experiments

## Additional information

**Competing financial interests:** The authors declare no competing financial interests.

