## [Peer Review File · Nature Communications]

Reviewers' Comments:

Reviewer #1 (Remarks to the Author)

The manuscript reports a novel form of microRNA sponge, designated a "zipper" an LNA antisense sequence that bridges from the 3' end of one microRNA to the 5' end of another, ultimately forming a long chain of immobilized microRNAs. The microRNAs miR-17, miR-221, and miR-222 were studied in human breast cancer cells. Upon transfection at 30 nM, the level of each target microRNA was reduced, wound healing on a slide was slowed, cyclin-dependent kinase inhibitor p27 was increased, and doxorubicin sensitivity was increased.

Nevertheless, measuring qPCR products with Sybr Green introduces ambiguity, relative to a specific microRNA fluorescent probe, thereby under-reporting the microRNA knockdown.

Considering the similarity of the miR-221 and miR-222 3' sequences, the anti-miR-221 zipper should have comparably knocked down miR-222, but the results say otherwise.

Appropriate statistical analysis and references were presented.

Confidence would be raised if luciferase vectors were used to study the activity of anti-microRNA zippers against individual microRNA target sequences identified in mRNA 3'UTR domains.

An RNAseq profile for at least one anti-microRNA zipper would throw light on the degree of off-target effects caused. At least a few experiments at 50 nM might show more definite efficacy of the anti-microRNA zipper approach, at least upon transfection.

Lastly, the presentation would be simpler if all results and figures were included in the main manuscript, rather than some in the supplementary data.

Reviewer #2 (Remarks to the Author)

A. Summary of key results

The manuscript under consideration is evaluating a new method to reduce the endogenous levels of miRNAs. The method relies on use of an antisense to the miRNA. But in this case, the antisense is generated such that it binds the 3' end of one miRNA and the 5' end of the next. In this way a miRNA/antisense zipper is generated. The authors show that this method can faithfully reduce the function of two miRNAs, miR-221 and miR-17. They claim that the small RNA zipper is at least as effective as current antagomirs. The zipper generated against miR-221 resulted in the sensitization of breast cancer cells to doxorubicin with higher affinity than when antisense miR-221 was used. While the technology has some novelty, there are some weaknesses with the experimental set-up and data presented that diminish the enthusiasm. Some of the major points are summarized below:

B. Originality and Interest:

Since many miRNAs belong to family that contain highly conserved members the authors should show that these zippers are capable of selectivity for one family member relative to another. The authors do show the zipper is specific for miR-221 and not another miRNA expressed from the same cluster, miR-222, which is important. However this does not address miRNAs that have strong homology between family members. miR-221 and miR-222, although family members, do not share the same level of homology as many other miRNA families, ie let-7. Thus, based on the proposed strategy it is not expected that the zipper would bind to and interfere with miR-222 since miR-221 and miR-222 are divergent in their 3' ends. The authors determine and show if this technology would work for miRNAs that share more homology to other family members.

Recent evidence suggests that many miRNAs have isomers. These isomers differ by the length of 1-3 nucleotides on either the 5' or 3' end of the molecule. It is not expected that the technology presented by the authors would account for this. Current antisense technologies are not limited by miRNA isomers.

The authors show function for the miR-221 zipper in preventing cell migration. Since antagomirs to miR-221 have already determined that preventing miR-221 activity can prevent migration this finding is not novel (le Sage C, Embo J, 2007; Garofalo M, Cancer Cell 2009; Stinson S, Science Signal 2011; Zheng C, Med Oncol. 2012; Quintavalle C, Oncogene 2012, etc). Similarly the authors show that the zipper that antagonizes miR-221 results in restoration of p27, a previously reported miR-221 target (Galardi, S J Biol Chem, 2009; le Sage C, Embo J, 2007; Fornari F, Oncogene 2008, etc). Thus the current technology is only being used to repeat what current tools have already shown and is incremental at best with regard to advancing biological and technological knowledge.

C. Data and Methodology:

There are no issues with the direct interpretation of the data presented or the methods used.

D. Appropriate use of statistics

Statistics are fine. Treatment of uncertainties needs to be addresses with regard to using this technology for isomers and specificity of this technology for miRNA families with a higher degree of similarity.

E. Conclusions

Overall the direct interpretation of the data is well supported. However the overall conclusion that this technology is superior to the use of antagomirs is not fully supported based on concerns mentioned above.

F. Suggested Improvements

Included targeting of isomers and family members with a high degree of similarity. Identify a novel target/function for the miRNA that is under evaluation.

G. References

No issues.

H. Clarity and context

No issues.

Reviewer #3 (Remarks to the Author)

This manuscript describes another approach to antagonizing a miRNA by designing antisense LNAs that bridge the 3'-end of one molecule to the 3'end to another and propose that they form a zipper that reduces the abundance of the miRNA. This is a clever idea. Knockdown is between 60-80% in different experiments, which isn't bad, but not necessarily better than other approaches. They claim that their approach is more effective than other anti-sense strategies. However, they don't provide convincing evidence that a zipper is formed, that the miRNA is degraded (rather than merely not PCR-amplified because of tight binding to the LNA oligos) or that the method is superior to existing antisense strategies (since other methods were not compared head to head or in dose response experiments). The latter is critical. They also do not discuss cytotoxicity (in fact in one immunoblot, it looks like there is reduced control protein suggesting possibly fewer surviving cells). They also claim that they uncovered a "novel" role of miR-221 in drug resistance, although there are multiple papers that show resistance to multiple drugs (although not to doxorubicin) when miR-221 is antagonized.

To reviewer #1

We thank the reviewer for the helpful suggestions and important questions as these additional experiments have strengthened the conclusions drawn.

Q1: “measuring qPCR products with Sybr Green introduces ambiguity, relative to a specific microRNA fluorescent probe, thereby under-reporting the microRNA knockdown”.

In addition to the SYBR Green approach (Figure 2B and 2C), Taqman probe approach was performed to further demonstrate the knockdown of miR-17 expression in 24h and 48h after transfection with miR-17 zipper into MDA-MB-231 cells (Supplemental Figure S1).

Q2: “Considering the similarity of the miR-221 and miR-222 3' sequences, the anti-miR-221 zipper should have comparably knocked down miR-222, but the results say otherwise.”

miR-221 and miR-222 share the same “seed” sequence but the rest of the sequence is different (as shown in Supplemental Figure S3). According to the structure of miR-221 zipper (Figure 2D), miR-221 zipper is not able to bind the 3' half sequence of miR-222, so it cannot connect miR-222 molecules to form a long chain. From our multiple experiments as shown in Figure 3C, 3D and Supplemental Figure S4, miR-222 did not show response to either low or high concentration of miR-221 zipper.

Q3: “Confidence would be raised if luciferase vectors were used to study the activity of anti-microRNA zippers against individual microRNA target sequences identified in mRNA 3'UTR domains”

To address this question, luciferase reporter constructs carrying either wild type p27 3'UTR or point mutated p27 3'UTR were tested in MDA-MB-231 cells with or without transfection with miR-221 zipper (Figure 6D). p27 is a target gene of both miR-221 and miR-222. Increased luciferase activity by miR-221 zipper to WT p27 3'UTR was observed (Figure 6E). However, the increased level of luciferase activity by miR-221 zipper was not as high as that by the mutated construct mainly due to the unaffected expression of miR-222 in miR-221 zipper-treated cells.

Q4: “At least a few experiments at 50 nM might show more definite efficacy of the anti-microRNA zipper approach, at least upon transfection.”

To address this question, we treated MDA-MB-231 cells with up to 50nM of the miR-221 zipper. As shown in Figure 2G and Supplemental Figure S5, 50nM did show higher efficiency to knockdown miR-221. Especially in 24h after transfection, the miR-221 level was suppressed ~90%.

Q5: “the presentation would be simpler if all results and figures were included in the main manuscript, rather than some in the supplementary data.”

Yes, we reorganized the figures, and put several important figures which were supplementary data in the previous version to the main manuscript in the current version, such as Figure 3E and 3F, Figure 5A and 5B. Since quite a few additional experiments were performed and new figures were generated, we still put a few data as supplementary figures to meet the figure number limit of the journal.

To reviewer #2

We thank the reviewer for the supportive remark indicating “the technology has some novelty”. We thank the reviewer for the helpful suggestions and important questions.

Q1: “Since many miRNAs belong to family that contain highly conserved members the authors should show that these zippers are capable of selectivity for one family member

relative to another. The authors do show the zipper is specific for miR-221 and not another miRNA expressed from the same cluster, miR-222, which is important. However this does not address miRNAs that have strong homology between family members. miR-221 and miR-222, although family members, do not share the same level of homology as many other miRNA families, ie let-7. Thus, based on the proposed strategy it is not expected that the zipper would bind to and interfere with miR-222 since miR-221 and miR-222 are divergent in their 3' ends. The authors determine and show if this technology would work for miRNAs that share more homology to other family members.”

To address this good question, firstly we applied the let-7a zipper to MDA-MB-231 cells, and determine the effects to the expression of let-7a, let-7b and let-7c. As shown in Figure 4A and 4B, only let-7a responded to the let-7a zipper, while let-7b and let-7c did not. In addition, a point mutated miR-17 zipper was designed and transfected into MDA-MB-231 cells as shown in Figure 4C. Two nucleotides mutation to the miR-17 zipper completely attenuated its function (Figure 4D). As a result, the miRNA zipper showed high sequence-specificity.

Q2: “Recent evidence suggests that many miRNAs have isomers. These isomers differ by the length of 1-3 nucleotides on either the 5' or 3' end of the molecule. It is not expected that the technology presented by the authors would account for this. Current antisense technologies are not limited by miRNA isomers.”

To address this important question, iso-miR-17 and iso-miR-221 was analyzed in the miR-17 zipper and miR-221 zipper treated cells, respectively. As seen in Figure 4E-4H, the miRNA zipper did show knockdown effects to the target miRNA and isomirs. The sequences of the two tested isomirs were from miRBase.

Q3: “The authors show function for the miR-221 zipper in preventing cell migration. Since antagomirs to miR-221 have already determined that preventing miR-221 activity can prevent migration this finding is not novel (le Sage C, Embo J, 2007; Garofalo M, Cancer Cell 2009; Stinson S, Science Signal 2011; Zheng C, Med Oncol. 2012; Quintavalle C, Oncogene 2012, etc). Similarly the authors show that the zipper that antagonizes miR-221 results in restoration of p27, a previously reported miR-221 target (Galardi, S J Biol Chem, 2009; le Sage C, Embo J, 2007; Fornari F, Oncogene 2008, etc).

Thus the current technology is only being used to repeat what current tools have already shown and is incremental at best with regard to advancing biological and technological knowledge.’

Yes, both the function of miR-221 in promoting cell migration and p27 as a target gene of miR-221 in breast cancer cells have been reported by our previous work and other groups’ work. Since this manuscript focused a novel approach to inhibit miRNA, we chose the well-demonstrated phenotypes of miR-221 to confirm the miR-221 zipper works well and has function in cells. In addition, by applying this technology a novel function of miR-221 in regulating doxorubicin resistance in breast cancer cells was confirmed as shown in Figure 5E-5H.

To reviewer #3

We thank the reviewer for stating “This is a clever idea. Knockdown is between 60-80% in different experiments, which isn't bad, but not necessarily better than other approaches.”. We thank the reviewer for the helpful suggestions and important questions.

Q1: “ they don't provide convincing evidence that the method is superior to existing antisense strategies.

To address this question, a head to head comparison between the miRNA zipper and existing antisense were performed in a dose response manner. As shown in Figure 2G and Supplemental Figure S5, both miRNA zipper method and existing antisense method can knockdown the target miRNA ~90% at most in 24h at concentration of 50nM. As such we stated in the manuscript that the zipper approach is at least as effective as the existing antisense approach in the case of miRNA knockdown and rescuing the expression of miRNA target genes. In terms of regulating dox-resistance of breast cancer cells, the miR-221 zipper showed higher efficiency than anti-miR-221 to improve the sensitivity to doxorubicin of breast cancer cells (Figure 5G and 5H).

Q2: “They also do not discuss cytotoxicity” of the miRNA zipper.

To address this question, different concentrations (0nM, 10nM, 30nM and 50nM) of miR-221 zippers were introduced into MDA-MB-231 cells. After 24h, Annexin V staining was performed to quantify the dead and apoptotic cells (Figure 4I). Quantitative analysis indicated ~5% of total dead and apoptotic cells in all tests. No significant difference was

found between control and miR-221 zipper treated cells (Figure 4J) indicating the nontoxicity of small RNA zipper to cells. H₂O₂ was used as a positive control to induce cell apoptosis (Supplemental Figure S6).

Q3: They also claim that they uncovered a "novel" role of miR-221 in drug resistance, although there are multiple papers that show resistance to multiple drugs (although not to doxorubicin) when miR-221 is antagonized"

Yes, miR-221 regulation of drug resistance has been reported by other groups and discussed in page 12 of the current manuscript. Although miR-221 regulating doxorubicin resistance has not been reported before, we are still careful in the current version not claiming it is a novel function of miR-221. Thanks again for the important comment.

Reviewers' Comments:

Reviewer #1 (Remarks to the Author)

The manuscript reports a novel form of microRNA sponge, designated a "zipper" an LNA antisense sequence that bridges from the 3' end of one microRNA to the 5' end of another, ultimately forming a long chain of immobilized microRNAs.

The microRNAs miR-17, miR-221, and miR-222 were studied in human MDA-MB-231 triple negative breast cancer cells. Upon transfection up to 50 nM, the level of each target microRNA was reduced, wound healing on a slide was slowed, cyclin-dependent kinase inhibitor p27 was increased, and doxorubicin sensitivity was increased.

In response to reviewer questions, the revised manuscript rigorously tested sequence specificity, and efficacy relative to ordinary antagomirs. The new version appropriately answered the reviewers' comments, and strengthened the original conclusions.

The revised manuscript is now suitable for publication.

Reviewer #2 (Remarks to the Author)

A. Summary of key results

The manuscript under consideration is a resubmission that is based on reducing the endogenous levels of miRNAs through the use of small RNA zippers. The authors have addressed some of the previous comments through discussion and new data. However, there are still some concerns with the new data and interpretation.

- It is not entirely clear how the authors envision the technology working on isomers that have more than 1nt. Based on the proposed technology, the oligos are generated to bind to the 3' end of one miRNA and the 5' end of another miRNA with a single nt linking the two. Thus, any additional nucleotides, beyond 1 nt, at the 5' or 3' end of the miRNA could impair binding. The cartoon in Figure 1 does not account for this.
- For all the figures not showing the effect of the control zipper (ie. 2B, 2E, 2G) it is not clear if the data are graphed relative to the effect of the negative control. If not, the data should be. In all cases the negative control should be tested and data graphed relative to the effect of the negative control. This is especially relevant for figure 2G where the higher doses of the oligo are tested. The authors do not show that the control, at the highest dose of 50 nM, does not alter the expression of miR-221.
- The northern blot data to support miR-17 knockdown is not convincing. qRT-PCR graphs argue that the miR-17 zipper knocksdown miR-17 levels by ~70%; however, the northern suggest otherwise. In fact, miR-17 zipper does not appear to reduced miR-17 at all. The levels of miR-17 look very similar between all lanes which contradicts the qRT data in figure 2B/C. The authors also claim that removing the 1-nt spacer diminished antagonizing of miR-17. Again, the Northern blot does not show that the wt miR-17 zipper has any activity at repressing miR-17 levels, thus changes when the mutated zipper is tested cannot be determined.
- For the let-7 study, the authors need to show that let-7a, let-7b and let-7c are expressed to similar levels in the cell line. Showing relative abundance does not support their finding that the zipper does not alter let-7b or let-7c levels. Let-7b and let-7c may be expressed at lower levels or not even be expressed in this cell line. In previous studies, let-7b and let-7c were 40-50% lower than let-7a in MB-231 cells. Let-7e however was shown to be expressed 2.5 fold more than let-7a.

Thus, qRT PCR of let-7e would be more telling of whether the zippers target family members with similar sequence. Even still, the authors should show the relative levels of the family members prior to selecting/showing family member expression in the presence of the zipper.

- In relation to the let-7 data, the full 5' end of both let-7b and let-7c should efficiently bind the zipper since the sequence is identical to let-7a. Additionally, the nucleotides that are divergent from let-7b change adenosine to guanine in both let-7b (two changes) and let-7c (a single A-G change). This ultimately would change an A-U base pair between the miRNA and the zipper to a G-U wobble pair, which would likely interact. Thus there should be some (albeit perhaps reduced) targeting of these two family members. If these miRNAs are expressed, based on the technology, they too should be interacting with the 3' end of the zipper and perhaps less robustly with the 5' end of the zipper precluding their activity somewhat. Without knowing the expression of let-7b and let-7c the data presented are not convincing based on the biology.

- Although the authors attempt to support the let-7 studies by mutating the miR-17 zipper, the authors mutated both the 5' and 3' end of the zipper. In this case, dual mutation (and mutations that do not support a wobble) will likely reduce silencing, as shown. To represent a similar situation to let-7 the authors should include a single mutation in the miR-17 zipper (either 5' or 3') that introduces a wobble. It is expected that a single wobble mutation would still lead to some silencing of miR-17 as explained above for the let-7.

- The isomers tested by the authors were only a single nucleotide longer than the miRNA. While the authors do show that these isomers are still silenced the question still remains as to whether or not isomers with more than 1 extra nt would also be silenced. Because there is only a 1 nt gap in the zipper, it is expected that more than 1 nt isomers would not be silenced. Thus, the data presented do not fully support the conclusion that the zippers will work on isomers. Also, the authors should show the relative level of the miR-17 (miR-221) isomers in the cell. Moreover, the isomer tested for miR-17 has a uracil added. The zipper has a guanine in this position. Thus, it is likely, based on G-U wobble base pairing that the interaction would occur. Silencing is not as robust when a mismatch is present (as shown for the miR-221 isomer). Again, suggesting that isomers with more than 1 nt would also not be silenced robustly or not at all.

- Data in figure 4H and 2G do not support each other. In 2G, 30 nM of miR-221 produced only a 60% knockdown while in 4H the authors show near complete knockdown. Data from both experiments were done identically based on methods/legends. The large discrepancy between the two knockdowns questions reproducibility of the data.

- Some figures have the X-axis labeled "relative expression" while some say "relative abundance" the authors need to clarify the difference between these axis titles.

- It is still not clear why the zipper approach is better than current antagomir technologies. Knockdown of the miRNAs is similar with both approaches. Thus, there does not appear to be any strong data supporting developing a new technology. If the response is not significantly better or toxicity more reduced than the current strategy the argument is not convincing.

B. Originality and Interest:

C. Data and Methodology:
Comments above.

D. Appropriate use of statistics

Statistics are fine. Treatment of uncertainties needs to be addressed with regard to using this technology for isomers and specificity of this technology for miRNA families with a higher degree of similarity. New data begin to address this but are not completely convincing

E. Conclusions

Still the overall conclusion that this technology is superior to the use of antagomirs is not fully supported. Similarly, some of the new data presented do not completely address the previous concerns.

F. Suggested Improvements

See "A" above for suggestions

G. References

No issues.

H. Clarity and context

No issues.

Reviewer #3 (Remarks to the Author)

The authors have adequately replied to the reviews. It would be helpful to have the manuscript edited for clearer and more precise use of English.

To reviewer #1

We thank the reviewer for the comment “the revised manuscript is now suitable for publication”.

To reviewer #3

We thank the reviewer for the comment “the authors have adequately replied to the reviewers”. We have had the manuscript language-edited for better use of English following the reviewer’s suggestion.

To Reviewer 2

We thank the reviewer for the comprehensive review and detailed comments regarding our miRNA zipper work. The comments are really helpful for us to improve the manuscript. To address the reviewer’s concerns, additional experiments have been performed. The revised figures and text have been updated. We believe the revised manuscript is much improved following the suggestions of the reviewers. We hope the current version provides satisfactory changes to address all the reviewer’s comments. We greatly appreciate your comments and help.

Q1: “IsomiRs with any additional nucleotides, beyond 1 nt, at the 5' or 3' end of the miRNA could impair binding. The cartoon in Figure 1 does not account for this.”

To answer this question, iso-miR-17 and iso-miR-221 with an extra 1 nt and 2 nts at the 3' end were tested for miR-17 zipper and miR-221 zipper, respectively. The iso-miR results have been updated in Figures 4E-4H. The experiment was repeated more than three times demonstrating the miRNA zipper did work on isomiRs with 1 or 2 extra nts at the 3' end, while the knockdown efficiency was a little bit variable between tested miR-17 and miR-221. In our opinion, miRNA zippers may work on isomiRs with additional nucleotides beyond 1 nt through a way as shown in the cartoon in supplemental Figure S6B. However, since isomiRs with 3 or more extra nts have not been tested yet, isomiR-sequence specific zipper (iso-miR zipper), following the design strategy shown in Figure 1, is strongly suggested rather than miR zipper to knockdown isomiRs, as discussed on page 8.

Q2: “For all the figures not showing the effect of the control zipper (ie. 2B, 2E, 2G) it is not clear if the data are graphed relative to the effect of the negative control. This is especially relevant for figure 2G where the higher doses of the oligo are tested. The authors do not show that the control, at the highest dose of 50 nM, does not alter the expression of miR-221.”

All the experiments were performed with a negative control, and all results were normalized to the negative control as mentioned in Results (page 5). In the current version, the effects of the control zipper on miR-17 and miR-221 were added as Supplemental Figure S1B and Supplemental Figure S2A. In addition, the effects of low dose (10 nM) to high dose (50 nM) of the control zipper on miR-221 abundance were shown in Figure 2H, which is complementary to Figure 2G.

Q3: “The northern blot data to support miR-17 knockdown is not convincing.”

To address this question, a quantitative analysis of the northern blot was performed. As indicated in updated Figure 3F, the bands intensity measurement showed miR-17 knockdown by the zipper at both 20 nM and 30 nM concentrations, while the mutated zipper (Δ zipper) worked at only the high concentration of 30 nM, and not at 20 nM. This indicated the reduced silencing effect of miR-17 Δ zipper, which is consistent with data from the miR-221 Δ zipper (Figure 3D).

Q4: “For the let-7 study, the authors need to show that let-7a, let-7b and let-7c are expressed to similar levels in the cell line. Showing relative abundance does not support their finding that the zipper does not alter let-7b or let-7c levels. The authors should show the relative levels of the family members prior to selecting/showing family member expression in the presence of the zipper.”

In consideration of effects of primer sequence on the DNA amplification efficiency, it may not be the best way to compare the expression levels of let-7a, b and c in one sample by real-time PCR analysis. So the absolute Ct values were shown in Supplemental Figure S5, which can reflect the relative abundance of let-7s in MDA-MB-231. Supplemental Figure S5 showed the similar Ct values of let-7a and let-7b (~22 cycles), while 4 more cycles for let-7c (~26

cycles), indicating the moderate expression of let-7a ,b and c in MDA-MB-231 cells, although let-7c has lower levels compared to let-7a and let-7b.

Q5: “In relation to the let-7 data, the full 5' end of both let-7b and let-7c should efficiently bind the zipper since the sequence is identical to let-7a. Additionally, the nucleotides that are divergent from let-7b change adenosine to guanine in both let-7b (two changes) and let-7c (a single A-G change). This ultimately would change an A-U base pair between the miRNA and the zipper to a G-U wobble pair, which would likely interact.”

This is a very important question. Yes, the two A-G changes in let-7b and one A-G change in let-7c did induce G-U wobble base-pairing. In order to enhance binding specificity, Locked Nucleic Acid (LNA™) nucleosides were applied to synthesize the miRNA zippers. Furthermore, the miRNA zippers are DNA oligos. We are not sure whether LNA™ G-T has wobble interaction as G-U before further validation. In addition, the sequence specificity of current antagomir technology has excluded the wobble interaction during miRNA knockdown.

In order to make this clearer, we tested the effects of the let-7a zipper on the abundance of let-7a, let-7b and let-7c several times. As shown in Figure 4B, let-7a zipper can knockdown let-7a efficiently, while it only has a minor effect on let-7b and let-7c with p values more than 0.05. The data did not support the wobble interaction between the let-7a zipper and let-7b and let-7c. In addition, following the reviewer’s suggestion, a single A-G mutation in the miR-17 zipper was induced to support a wobble interaction (miR-17-zipper mu2) as shown in Figure 4C and 4D. miR-17 can be efficiently knocked down by miR-17 zipper, but not at all by miR-17-zipper mu1. miR-17-zipper mu2 could decrease the miR-17 level a little bit, but the p value did not show statistical significance.

Q6: “Although the authors attempt to support the let-7 studies by mutating the miR-17 zipper, the authors mutated both the 5' and 3' end of the zipper. In this case, dual mutation (and mutations that do not support a wobble) will likely reduce silencing, as shown. To represent a similar situation to let-7 the authors should include a single mutation in the miR-17 zipper (either 5' or 3') that introduces a wobble.”

Same answer as to Q5.

Q7: “The isomers tested by the authors were only a single nucleotide longer than the miRNA. While the authors do show that these isomers are still silenced the question still

remains as to whether or not isomers with more than 1 extra nt would also be silenced. The authors should show the relative level of the isomers in the cell.”

Same answer as to Q1, iso-miR-17 and iso-miR-221 with either 1 or 2 extra nt were tested and the results are shown in Figures 4E-4H.

In addition, the relative levels of the iso-miRs were indicated in Supplemental Figure S6A with absolute Ct values from real-time PCR analysis.

Q8: “Data in figure 4H and 2G so not support each other. In 2G, 30 nM of miR-221 produced only a 60% knockdown while in 4H the authors show near complete knockdown. Data from both experiments were done identically based on methods/legends. The large discrepancy between the two knockdowns questions reproducibility of the data.”

All experiments were repeated more than 3 times in triplicates. As mentioned in the abstract and main text, “Two miRNAs, miR-221 and miR-17, were tested in human breast cancer cell lines demonstrating the 70~90% knockdown of miRNA levels by 30-50 nM small RNA zippers”. The knockdown effect may be a bit variable from different repeats due to variations from cell conditions and/or transfection efficiency.

Q9: “Some figures have the X-axis labeled "relative expression" while some say "relative abundance" the authors need to clarify the difference between these axis titles.”

We have updated all the axis-labelings. “miRNA expression” was used to label Figure 5E and 5F, where the miRNA levels represented the transcriptional levels. When the cells were treated with miRNA zippers. “miRNA abundance” was used to label Y-axis.

Q10: “It is still not clear why the zipper approach is better than current antagomir technologies.”

This work demonstrated a novel approach to knockdown miRNAs as well as other small RNAs. A head to head comparison between the miRNA zipper and existing antisense was performed in a dose response manner. As shown in Figure 2G and Supplemental Figure S4, both the miRNA zipper method and the existing antisense method can knockdown the target miRNA ~90% at most in 24h at a concentration of 50 nM. As such we stated in the manuscript that “a head to head comparison between the miRNA zipper and antisense inhibitor indicated similar efficiency to knockdown the target miRNA expression” on page 11. In terms of

regulating dox-resistance of breast cancer cells, the miR-221 zipper showed higher efficiency than anti-miR-221 to improve the sensitivity to doxorubicin of breast cancer cells (Figures 5G and 5H).

Reviewers' Comments:

Reviewer #2 (Remarks to the Author)

The revised manuscript, "Small RNA zippers: lock miRNA molecules and block miRNA function in mammalian cells" has been substantially revised. This submission includes new experimental data and discussion that fully supports the use of miRNA zippers for silencing miRNA activity. The authors have addressed all of the previous comments. Thus, the manuscript is now acceptable for publication as is.